# Electrospinning vs. Electro-Assisted Solution Blow Spinning for Fabrication of Fibrous Scaffolds for Tissue Engineering

**DOI:** 10.3390/polym14235254

**Published:** 2022-12-01

**Authors:** Tatiana S. Demina, Evgeniy N. Bolbasov, Maria A. Peshkova, Yuri M. Efremov, Polina Y. Bikmulina, Aisylu V. Birdibekova, Tatiana N. Popyrina, Nastasia V. Kosheleva, Sergei I. Tverdokhlebov, Peter S. Timashev, Tatiana A. Akopova

**Affiliations:** 1World-Class Research Center “Digital Biodesign and Personalized Healthcare”, Sechenov First Moscow State Medical University, 8-2 Trubetskaya St., 119991 Moscow, Russia; 2Weinberg Research Center, School of Nuclear Science & Engineering, National Research Tomsk Polytechnic University, 30 Lenin Ave, 634050 Tomsk, Russia; 3Institute for Regenerative Medicine, Sechenov First Moscow State Medical University, 8-2 Trubetskaya St., 119991 Moscow, Russia; 4Laboratory of Clinical Smart Nanotechnologies, Sechenov First Moscow State Medical University, 8-2 Trubetskaya St., 119991 Moscow, Russia; 5Enikolopov Institute of Synthetic Polymeric Materials, Russian Academy of Sciences, 70 Profsoyuznaya St., 117393 Moscow, Russia; 6FSBSI Institute of General Pathology and Pathophysiology, 8 Baltiyskaya Ulitsa, 125315 Moscow, Russia; 7Chemistry Department, Lomonosov Moscow State University, Leninskiye Gory 1-3, 119991 Moscow, Russia

**Keywords:** polylactide, non-woven mats, electrospinning, tissue engineering, solution blow spinning, cell growth, biopolymers

## Abstract

Biodegradable polymeric fibrous non-woven materials are widely used type of scaffolds for tissue engineering. Their morphology and properties could be controlled by composition and fabrication technology. This work is aimed at development of fibrous scaffolds from a multicomponent polymeric system containing biodegradable synthetic (polylactide, polycaprolactone) and natural (gelatin, chitosan) components using different methods of non-woven mats fabrication: electrospinning and electro-assisted solution blow spinning. The effect of the fabrication technique of the fibrous materials onto their morphology and properties, including the ability to support adhesion and growth of cells, was evaluated. The mats fabricated using electrospinning technology consist of randomly oriented monofilament fibers, while application of solution blow spinning gave a rise to chaotically arranged multifilament fibers. Cytocompatibility of all fabricated fibrous mats was confirmed using in vitro analysis of metabolic activity, proliferative capacity and morphology of NIH 3T3 cell line. Live/Dead assay revealed the formation of the highest number of cell–cell contacts in the case of multifilament sample formed by electro-assisted solution blow spinning technology.

## 1. Introduction

Non-woven mats are widely proposed as scaffolds for tissue engineering since they have high porosity and structural similarity to the morphology of native living tissues [1,2]. Electrospinning (ES) technology is considered to be one of the most perspective, reliable, technological and versatile approaches for the fabrication of non-woven materials for a wide range of applications [3,4,5]. This technology relies on the traveling of electrically charged polymeric solution or melt to a collector accompanied by stretching of the polymeric jet into nano/microfibers, while facilitating fiber solidification. The structure of the produced mats could be significantly varied in terms of orientation, fiber diameter and morphology as a function of processing conditions, such as solution/melt composition, set-up characteristics [6,7,8]. Synthetic polymers (polylactide, polycaprolactone, etc.) could be spine starting from melt or non-aqueous solutions [9]. Water-soluble polymers of both synthetic (polyethylene oxide) and natural (proteins, polysaccharides) origins could require a better control of solution characteristics, processing conditions and cross-linking of the materials [10,11,12]. The formed non-woven mats consist of monofilaments which are mostly monolithic, but their structure could be made more complex (core/shell, porous, etc.) via modification of the processing conditions [13,14].

Solution blow spinning (SBS) is a promising alternative to the electrospinning method [15,16]. SBS is based on the driving of polymeric solution by a gas flow. In contrast to ES, SBS normally is less strict in terms of process requirements, i.e., polymeric solution does not need to be electroconductive. A low coefficient of fiber transfer from the injector to the substrate is the main limitation of the SBS technique. To better control the spinning process, SBS also could be realized using an electrostatic field as an additional driving force for the polymeric solution deposition [16]. Such electro-assisted SBS (EA-SBS) or electro-blown spinning combines the advantages of both ES and SBS technologies. The fundamental basis and modeling of the EA-SBS processing is intensively developing [17,18]. This technology is used for fabrication of fire protection materials [19], nanofibrous electret filters [20], membranes for distillation [21], superabsorbents [22], etc. There is a limited number of works on application of EA-SBS technique for biomaterial fabrication, such as bioactive hydroxyapatite fibers [23], antimicrobial nonwovens [24] and scaffolds for tissue engineering [25]. One of the most important advantages of SBS and EA-SBS in terms of the fabrication of scaffolds for tissue engineering is the multifilament structure of the produced fibers. Such a type of morphology could be superior to the monofilament in terms of cell adhesion and proliferation. Thus, multifilament knitted fibrous materials were shown to be more favorable for cell growth than monofilament ones [26,27]. Comparison of growth of bone marrow-derived stromal cells onto polylactide mats produced via either ES or SBS technology showed better results in the case of the sample made using SBS [28].

Fabrication of non-woven scaffolds for tissue engineering requires not only control over the mat’s morphology but fiber composition as well. One type of polymers is not enough to provide optimal characteristics needed for “ideal” scaffolds, but a combination of various polymers could compromise the processability of a composite system. To overcome the thermodynamic incompatibility of biodegradable polymers widely proposed for fabrication of scaffolds, i.e., the synthetic (polylactide, polycaprolactone) and the natural (gelatin, chitosan), a multicomponent system containing a fraction of graft-copolymers was synthesized [29]. The polyester part of the system allowed for keeping the processability form easily evaporated in organic aprotic solvents, while natural components provided biocompatibility to mats fabricated from this system via ES from chloroform.

This work was aimed at the study of processability of the multicomponent system via ES and EA-SBS technologies as well as evaluation of structure and biocompatibility of the fabricated fibrous materials.

## 2. Materials and Methods

### 2.1. Materials

Polylactide (PLA) (Natureworks 4043D, Minnetonka, MN, USA) with molecular weight (Mw) of 100 kDa; polycaprolactone (PCL) with Mw of 67 kDa and gelatin (Chimmed, Moscow, Russia) were used as received. Chitosan with Mw of 80 kDa and degree of deacetylation of 0.89 as well as oligo(l-lactide) with Mw of 5 kDa from l-lactic acids (Panreac, Barcelona, Spain) were synthesized in Enikolopov Institute of Synthetic Polymeric Materials. Multicomponent copolymer-contained system based on polycaprolactone/poly(l-lactide)/oligo(l-lactide)/chitosan/gelatin (PPCOG) was obtained via mechanochemical approach and characterized as reported earlier [29]. Hexafluoroisopropanol (HFIP) was purchased from “Chemmed” (Moscow, Russia) and used as received.

### 2.2. Fabrication and Physical-Chemical Properties of the Fibrous Materials

For the scaffold fabrication PPCOG was dissolved in HFIP in a closed glass vial at 50 °C to obtain the viscous opalescent solution with the concentration of 14 wt.%. The viscosity of the solution measured using viscometer SV-10 (AND, Tokyo, Japan) was (127 ± 4) × 10^−3^ Pa × sec. Electrical conductivity evaluated with the aim of inoLab Cond 7319 conductometer with a TetraCon 325 measuring cell (WTW, Weilheim, Germany) was 38.2 ± 0.6 µS/cm.

Non-woven fibrous materials were fabricated via electrospinning or solution blow spinning technology using either commercial NANON-01A electrospinning setup (MECC Co., Ltd., Fukuoka, Japan) or the SBS laboratory set-up described earlier [30,31]. In the case of ES method an aluminum cylinder with a diameter of 200 mm and a length of 100 mm was used to collect the fibers. The following parameters were used for the material fabrication: applied voltage of 22 kV, injector-to-collector distance of 70 mm, solution feed rate of 5 mL/h, collector rotation rate of 200 rpm. A 22 G needle was used as injector. In the case of SBS technique an aluminum cylinder with a diameter of 200 mm and a length of 100 mm was used to collect the fibers. The following technological parameters were used: an air pressure of 0.35 MPa, a flow rate of the polymer solution of 80 mL/h, a nozzle diameter for supplying the polymeric solution of 0.8 mm, a nozzle diameter for supplying compressed air of 1.7 mm, and a distance from the nozzle to the collector of 40 cm. To realize the EA-SBS mode the additional DC-discharge electric field generated by BQ040R250 (XP Power, Rungis Cede, France) at the voltage of 22 kV was applied. To remove the residual solvent, the fabricated materials were stored in a VD 115 vacuum furnace (Binder, Tuttlingen, Germany) at a temperature of 50 °C and a 0.1 Pa pressure for 24 h.

Structure and surface morphology of the materials were evaluated using scanning electron microscopy (SEM) with an aim of Phenom ProX (Thermo Fisher Scientific, Waltham, MA, USA) at 10–15 kV. The SEM images were processed using ImageJ software (version 1.52) to calculate the fiber diameter’s size distribution and local porosity [32,33].

The surface contact angles values of the fibrous materials were measured as a function of time using sessile drops of distilled water (mQ). The measurements were carried out using the Acam-MSC01 (Apex Instruments, Kolkata, India) within 33 min after a water drop set on different areas of the fibrous materials. The contact angles are reported as the average of at least three measurements; the standard deviation for contact angles was ±1 degree. To check the presence of natural components at the material surface the mats were stained with fluorescein isothiocyanate (FITC) (Sigma-Aldrich, St. Louis, MO, USA) and observed using fluorescent microscopy as described previously [29]. The water retaining capacity and swelling ratio of the fibrous materials was studied in mQ water. For this purpose, the pre-weighted samples (5 × 5 mm) were immersed in 10 mL of the water for 2 h. Before the weighting of the samples to calculate their water retention capacity, the surface water was removed with a filter paper from the mats. As the second option to calculate the material swelling, the samples before the weighting were manually pressed to remove the water retained within the pores. The results of water retention and swelling experiments are presented as average ± SD.

Mechanical properties of the fibrous samples were tested in dry and wet states using Mach-1 v500csst Micromechanical Testing System (Biomomentum Inc., Laval, QC, Canada). Testing of the wet samples was conducted after their incubation in mQ water for 2 days during constant lateral stirring at 200–300 rpm at RT. The tensile strength, elongation at break and elasticity modulus (Young’s modulus) were measured during uniaxial tension of at least three dog-bone shape cut samples (13 × 5 mm gauge dimensions, the thickness was 0.17 ± 0.1 mm for EA-SBS and 0.19 ± 0.1 mm for ES samples, respectively) and reported as mean ± standard deviation values.

### 2.3. In Vitro Cell Biocompatibility of Non-Woven Fibrous Mats

#### 2.3.1. Cell Culture

The fibrous mats cytocompatibility was analyzed using NIH 3T3 cells (mouse embryonic fibroblasts cell line). The culture medium consisted of Dulbecco’s Modified Eagle’s Medium (DMEM)/F12 supplemented with L-glutamine (1:1, Biolot, St. Petersburg, Russia), 10% fetal calf serum (HyClone, Logan, UT, USA), and gentamycin (50 μg/mL, Paneco, Moscow, Russia). Cell morphology was routinely checked with a phase-contrast microscope Primovert (Carl Zeiss, München, Germany).

#### 2.3.2. Material Cytocompatibility via Extract Test

Extracts of the samples were prepared via incubation of a 1 cm^2^ piece of each sample in 1 mL of culture media for 24 h at 37 °C. Serial dilutions of the four extracts and sodium dodecyl sulphate (SDS) dilutions as a positive control were performed and added in triplicate to NIH 3T3 cells seeded in 96-well plates. Cells were incubated for 24 h at 37 °C in 5% CO_2_. For the assessment of the non-woven mats extracts’ cytotoxicity, AlamarBlue cell viability reagent (Invitrogen, Waltham, MA, USA) was used. First, the media were replaced with the reagent solution according to the manufacturer’s instructions and were incubated for 2 h at 37 °C in 5% CO_2_. Then, the fluorescence of the viable cells was quantified using a spectrofluorometer Victor Nivo (PerkinElmer, Waltham, MA, USA) at a 530 nm excitation wavelength and a 590 nm emission wavelength. The Quant-iT PicoGreen kit (Invitrogen, Waltham, MA, USA) was used for the quantification of the DNA amount in the same samples to confirm the results obtained with the AlamarBlue assay. The AlamarBlue assay plates were washed thrice with sterile phosphate buffer saline (PBS) and filled with ultrapure H_2_O (dH_2_O). The plates underwent 3 freeze-thaw cycles to destroy the cell membranes and release intracellular DNA and then were operated, according to the manufacturer’s instructions. DNA fluorescence intensity was detected using a spectrofluorometer (Victor Nivo) at a 480 nm excitation wavelength and a 520 nm emission wavelength.

#### 2.3.3. Contact Cytocompatibility of the Fibrous Materials

For further cytotoxicity testing, contact culturing on the samples was performed. Briefly, NIH 3T3 cells were cultured on a 1 cm^2^ piece of the non-woven fibrous mats for 48 h (10^5^ cells per sample). For negative and positive controls, the cells were cultured in the two wells of a 24-well culture plate (10^5^ cells per well) in a complete culture medium and SDS (1 mg/mL) respectively. Then the metabolic and proliferative cell activities, as well as the biomaterials’ cytotoxicity, were assessed.

The non-woven fibrous mats’ cytotoxicity was assessed via LDH Assay using a Pierce LDH Cytotoxicity Assay Kit (Thermo Fisher Scientific, USA). Briefly, the medium from each well was transferred in triplicate to a new 96-well plate, and the reagent mix from the kit was added to each well, according to the manufacturer’s instructions. The plates were incubated for 30 min at 37 °C, the stop solution was added to each well, and the plates were read twice using a spectrofluorometer Victor Nivo at 495 nm and 625 nm excitation wavelength. The cells’ metabolic activity was assessed via AlamarBlue Assay as described before. Briefly, the media were replaced with the reagent solution according to the manufacturer’s instructions, and the cell-seeded constructs, as well as the two controls, were incubated for 2 h at 37 °C in 5% CO_2_. Then, the solution from each well was transferred in triplicate to a new 96-well plate, and the fluorescence was quantified using a spectrofluorometer Victor Nivo at 530 nm excitation wavelength and 590 nm emission wavelength. The cells’ proliferative activity was assessed via PicoGreen Assay as described before. Briefly, the samples and the control wells were thoroughly washed with sterile PBS and filled with dH_2_O. The plates underwent 3 freeze-thaw cycles to destroy cell membranes and release intracellular DNA. Then, the solution from each well was transferred in triplicate to a new 96-well plate and operated, according to the manufacturer’s instructions. DNA fluorescence intensity was detected using a spectrofluorometer (Victor Nivo) at 480 nm excitation wavelength and 520 nm emission wavelength.

Contact culturing on the samples was equally performed to assess cell viability via a Live/Dead assay. Briefly, NIH 3T3 cells were cultured on 1 cm^2^ piece of the non-woven fibrous mats for 48 h (10^5^ cells per sample) and then stained for 30 min at 37 °C in 5% CO_2_. Live cells were stained with Calcein-AM (Sigma-Aldrich, St. Louis, MO, USA); dead cells were stained with Propidium Iodide (Thermofisher, Waltham, MA, USA). Then, the samples were thoroughly washed with a complete growth medium, and their visualization was performed on EVOS M5000 Imaging System (Thermo Fisher Scientific, USA).

## 3. Results

### 3.1. Morphology and Physical-Chemical Properties of the Fibrous Materials

#### 3.1.1. SEM Microscopy of the Fibrous Materials

A preliminary screening of the processability of the multicomponent polymeric system via ES, SBS and EA-SBS techniques showed that defect-free mats could be fabricated using ES and EA-SBS approaches (Figure 1). An application of inert gas as the only driving force (SBS method) was not enough to fabricate fibrous mats. The mats fabricated via ES technology consist of randomly oriented monofilament fibers, while EA-SBS gave a rise to chaotically arranged multifilament fibers. Analysis of the SEM images showed that the ES allows for fabricating the fibers with a narrower diameter size distribution than those obtained in the course of EA-SBS technique (Figure 1).

Table 1 summarizes the results of the local porosity image analysis and experiments on water retention capacity and swelling of the fibrous mats. The EA-SBS-formed material has a higher porosity and, therefore, is able to retain more water than electrospun sample. The material swelling ratios were similar for both samples, since this characteristic refers to polymeric composition itself.

#### 3.1.2. Surface Properties of the Fibrous Materials

The fibrous material made via the ES approach had a lower wettability contact angle in comparison with that made using EA-SBS technique: 89 ± 1 and 102 ± 1 degrees, respectively (Figure 2). The contact angles decreased in time faster in the case of material fabricated via ES technique. The lower value of the initial contact angle and its faster decrease in time could be explained by the different morphology of the electrospun fibers. The EA-SBS material consists of fibers having a higher surface roughness and unevenness in diameter compared to ES, which could provoke trapping more air, causing more water repulsion [34].

Due to hygroscopic components within the PPCOG system, such as chitosan and gelatin, which could affect the size of water droplets in a course of contact angle measurements, the presence of these components within the surface layer of the fibrous materials was confirmed using fluorescein microscopy (Appendix A).

The samples were also mechanically tested not only in a dry state but in a wet state to better represent the material behavior within the cell culture conditions. Macrophotos of the samples at the beginning and at the end of the deformation are shown in Figure 3, while obtained results of the mechanical testing are summarized in Table 2.

ES samples had higher elastic modulus, tensile strength and elongation at break in the dry state. Incubation of the samples in water before the testing caused swelling of the hydrophilic biopolymers which led to some degree of softening (a decrease in elastic modulus and tensile strength), at the same time an increase in elongation at break was observed for the EA-SBS sample. Overall, the samples demonstrated mechanical properties expected for similar polymer systems and are in the range of those with connective tissues [29,35].

### 3.2. Cytocompatibility of the Non-Woven Mats

A variety of methods was used in this study to assess the biomaterials’ cytotoxicity, since it is generally recommended that such analysis should be performed using a combination of at least three methods based on different cell properties, e.g., the metabolic activity, proliferative capacity and morphology [36]. The AlamarBlue assay has demonstrated that neither of the samples’ extracts impaired the metabolic activity of NIH 3T3 cells, while SDS exposure led to cell death (IC50 ≈ 0.03 mg/mL) (Figure 4. The PicoGreen assay has shown that none of the extracts impaired the proliferative activity of NIH 3T3 cells either in contrast to SDS. While sample extracts-treated cells demonstrated high viability (≈80–100%) and a DNA count ranging from 450 to 700 ng/mL, SDS-treated cells showed no metabolic or proliferative activity at SDS concentration >0.03 mg/mL.

The NIH 3T3 cells cultured on the samples’ surface for three days demonstrated good metabolic and proliferative activity confirmed by AlamarBlue and PicoGreen assays (Figure 5b,c). However, the highest viability and proliferative activity were observed for the cells cultured on the sample formed by EA-SBS (marked with an asterisk) (Figure 5b,c). Moreover, this sample was the least cytotoxic to the cultured cells (Figure 5a).

Live/Dead assay revealed no contact cytotoxicity, since after 48 h in culture a high number of green viable cells could be observed on all scaffolds with only a few dead red cells (Figure 6). Images were analyzed using imaging software ImageJ (NIH, Bethesda, MD, USA). The background noise was removed, and gamma multiplication was applied to highlight bright areas and to suppress the darkest areas. The particle analysis revealed 5% of dead cells in the electrospun sample, and 4% of dead cells in the sample formed by EA-SBS. The cells evenly spread within the scaffolds, forming a considerable number of cell–cell contacts; however, the number of cell–cell contacts was the highest for the EA-SBS sample (Figure 6b).

## 4. Discussion

The difference in the fibrous materials morphology is obviously caused by various mechanism of the fiber formation. The process of the material forming by electrospinning is characterized by a high concentration of the like electric charges in the spinning solution, which is due to a relatively low flow rate of the spinning solution through the injection nozzle. Under the action of electrical forces, the jet of the polymer solution becomes transversely unstable and flattens [37]. After turning across the lines of force of the external electric field, the jet splits along its axis into two jets, approximately equal in volume, each of which is capable of splitting again. This process is repeated and continues until the capillary pressure on the surface of the jets compensates for the pressure of electric forces or the jet turns into a solid fiber upon evaporation of the solvent. Thus, the multiple process of splitting the jet under the action of electrical forces contributes to a decrease in the diameter of the generated fibers and provides a uniform smooth fiber surface.

In the case of the EA-SBS method, the consumption of the spinning solution through the injection nozzles is more than 15 times greater than that by the ES method. The strength of the electric field is insufficient for effective splitting of the spinning solution jet, and the main work on accelerating and stretching the spinning solution jet is performed by a non-isothermal high-speed gas flow. In this case, the effective stretching of the spinning solution jet occurs at a small distance (≈100 ÷ 150 µm) from the injection nozzle [16]. The stretching of the polymer jet causes a decrease in its diameter by a factor of approximately 100 compared to the diameter at the outlet of the injection nozzle. In this case, the speed of the polymer jet significantly increases and is approximately 23% of the speed of the air surrounding the jet. Reducing the diameter of the polymer jet is accompanied by intensive solvent evaporation, as a result, the viscosity of the polymer jet increases. Then, the polymer solution jet enters the gas flow zone, which is characterized by a high turbulence, where it is bent and entangled. Due to a high viscosity of the spinning solution in the turbulent region, the effective stretching of the jet and the disintegration of the jet into smaller jets practically does not occur. However, the probability of jet entanglement increases, which leads to the formation of fibers of complex morphology having a mainly micron size. In the case of EA-SBS technique, the electric field contributes to the settling of the fibers on the collector, reducing the loss of fibers and increasing the fiber productivity.

The structure and properties of the produced fibrous materials are in a good agreement with the main processing features of each technology. EA-SBS technology allows for generating the mat consisting of micro-sized multifilament fibers and having a higher porosity than the materials made via the ES method. Mechanical properties of the fibrous mats formed by the ES method are higher than that of scaffolds formed by the EA-SBS method (Table 2). Since the ES scaffolds are formed mainly by submicron fibers, while the average fiber diameter in EA-SBS scaffolds reaches tens of microns, the specific load onto each fiber is less for scaffolds formed by the ES method, which determines their greater absolute strength. Moreover, SEM images (Figure 1) show that the electrospin fibers have a regular cylindrical shape, while the EA-SBS-formed fibers have a more complex morphology and macrodefects in the form of drops. Therefore, a more perfect fiber structure of ES scaffolds contributes to their better mechanical properties.

The fiber diameter and material architecture are one of the most important characteristics of the scaffolds for tissue engineering. The scaffolds made of nanosized fibers promote cell proliferation on the material surface, while the cell migration within the material volume is limited [38]. On the contrary, the scaffold made on microfibers allows the mammalian cells to migrate and proliferate within the scaffold [39] as well as to stimulate the bone tissue regeneration [40]. In a frame of our work, the mean size of the fibers obtained by ES was in the submicron range (0.85 ± 0.38 μm), which was considered to be optimal for cell attachment and proliferation [32]. Such a type of submicron electrospun fiber superiority for cell growth is explained by an enlarged surface area and, thus, protein binding. The multifilament fibers produced by EA-SBS have the bigger mean fiber diameter size (13.70 ± 13.03 μm) but a rough surface morphology, which could provide a high surface area for protein binding. Indeed, the highest metabolic and proliferative activity was found for cells cultured on the fibrous material produced by the AE-SBS technique (Figure 5). The highest number of cell–cell contacts visible in the Live/Dead assay images supports the better cell attachment and growth of this material as well (Figure 6). The improved ability of the EA-SBS samples to support cell growth could be explained by easier medium circulation within the microfibrous scaffold having the bigger fiber diameter size and porosity. As a second option the multifilament surface roughness at the micron and submicron scales could be also favorable for cell cultivation [41].

Another factor affecting cell attachment is surface wettability. The cell adhesion is higher at the hydrophilic surfaces, which is also related to a protein adsorption [41]. The contact angle of wettability was lower in the case of the electrospun fiber material, but both types of the fibrous materials tend to show a decrease in contact angle in time, which is related to a swelling of the material containing natural hygroscopic components at the surface layer. As seen in Figure 2, within half an hour after a drop set the contact angles of wettability for both samples were below 80 °C, which is shown to be favorable for fibroblast adhesion and growth [42]. The high surface hydrophilicity promotes cell adhesion within the first hours of cell cultivation, but a too hydrophilic surface is not such an advantage in the long run, which gives the optimal contact angle in a range of 20–80 degrees. The cell growth is obviously a result of several material characteristics, such as chemical composition, wettability and morphology. Within our research the main variable material feature was morphology, which changed from mono- to multifilament as a function of spinning technology. The EA-SBS led to formation of multifilament fibrous materials providing higher cell attachment and growth than the ES-spun monofilament.

## 5. Conclusions

Electro-assisted solution blow spinning technique allows for fabricating non-woven fibrous materials consisting of multifilament fibers in contrast to monofilament electrospun fibers. Using the same concentration of the spinning solution the EA-SBS led to formation of fibers with a bigger mean diameter size than those produced by ES. The diameter size distribution was wider in the case of using the EA-SBS technique. The multifilament structure of the fibrous materials produced via EA-SBS provided better mammalian cell adhesion and growth than electrospun fibers. This phenomenon could be caused by a rougher multifilament fiber surface. The found superiority of multifilament fibrous materials in contrast to the monofilament in terms of cell attachment and proliferation makes them a highly promising type of scaffold for tissue engineering.

## Figures and Tables

**Figure 1 polymers-14-05254-f001:**
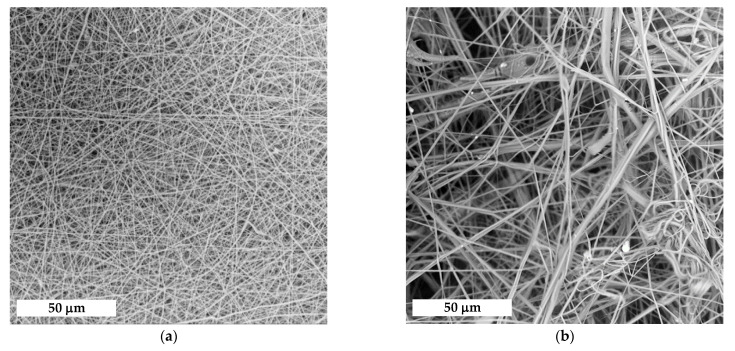
(**a**,**b**) SEM images and (**c**,**d**) histograms of the fiber diameter distribution for the fibrous materials made using (**a**,**c**) ES or (**b**,**d**) EA-SBS technique.

**Figure 2 polymers-14-05254-f002:**
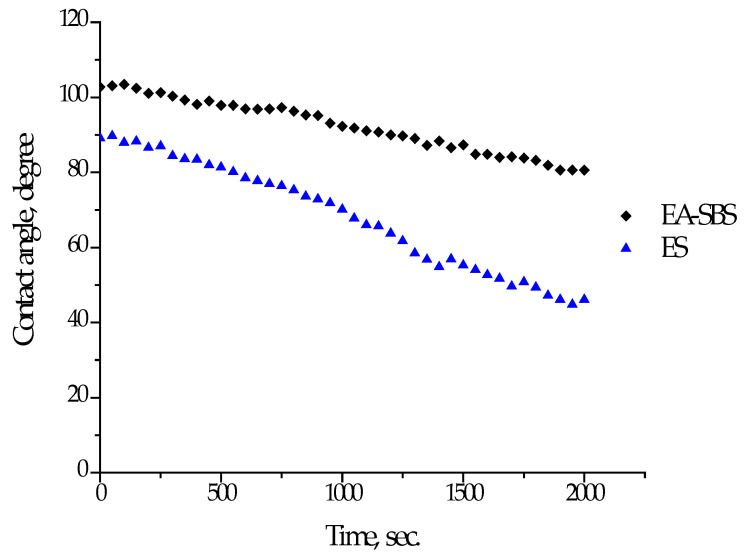
Dependence of contact angle of wettability of materials fabricated using ES or EA-SBS technique.

**Figure 3 polymers-14-05254-f003:**
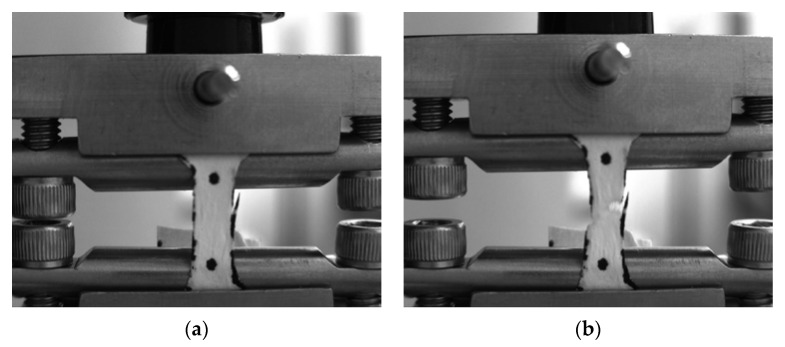
Photos of (**a**,**b**,**e**,**f**) EA-SBS and (**c**,**d**,**g**,**h**) ES samples in dry (**a**–**d**) and wet states (**e**–**h**) at the beginning and at the end of the mechanical testing.

**Figure 4 polymers-14-05254-f004:**
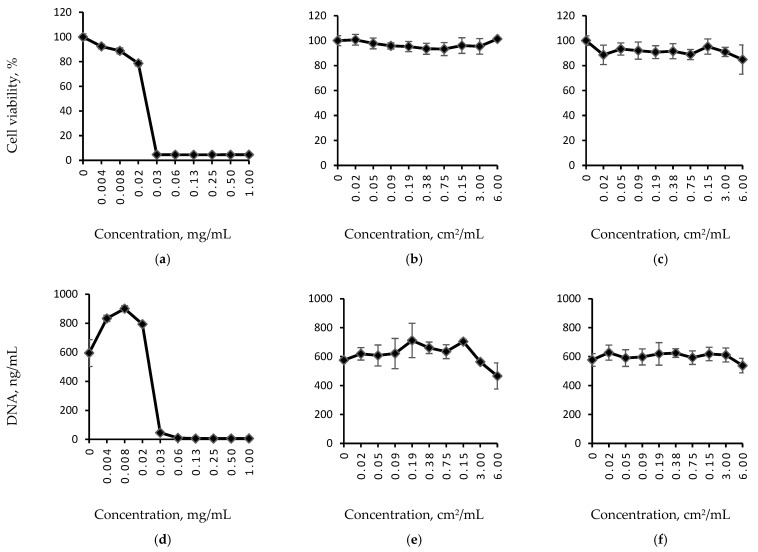
Relative cell viability (**a**–**c**) and DNA quantity (**d**–**f**) curves for sodium dodecyl sulphate (SDS) serial dilutions (**a**,**d**), and extracts of fibrous samples formed via (**b**,**e**) ES or (**c**,**f**) EA-SBS technique. SDS is used as a positive control.

**Figure 5 polymers-14-05254-f005:**
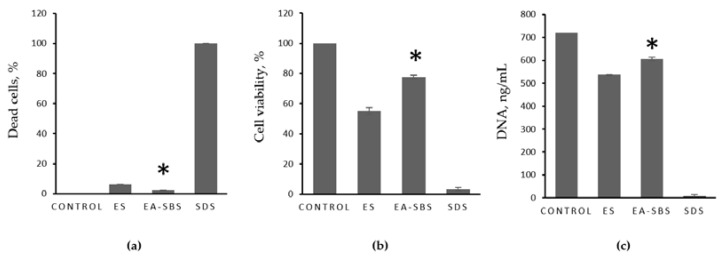
Relative biomaterials’ cytotoxicity (**a**), cell viability (**b**), and DNA quantity (**c**). 2D culture of NIH 3T3 cells in culture medium is used as a negative control, 2D culture of NIH 3T3 cells in 1 mg/mL sodium dodecyl sulphate (SDS) is used as a positive control. The sample formed by EA-SBS is marked with an asterisk.

**Figure 6 polymers-14-05254-f006:**
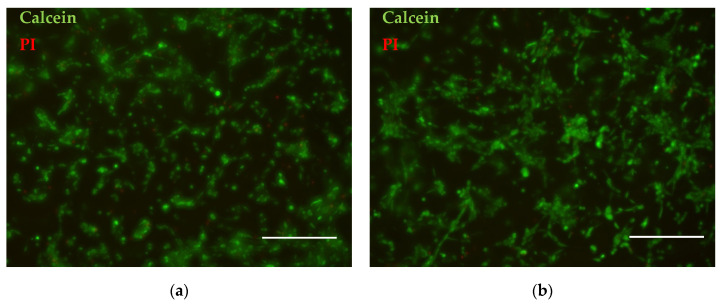
Live/Dead assay of cells cultured at fibrous materials fabricated using (**a**) ES or (**b**) EA-SBS technology. Live cells are stained with calcein and shown as green, dead cells are stained with Propidium Iodide (PI) and shown red. The scale bar is 250 μm.

**Table 1 polymers-14-05254-t001:** Effect of the fabrication technique on the fibrous materials morphology.

Fabrication Technology	Size of the Fiber Diameter, μm	Local Porosity, %	Water Retention Capacity, wt.%	Swelling, wt.%
ES	0.85 ± 0.38	37.9 ± 2.6	301 ± 39	178 ± 35
EA-SBS	13.70 ± 13.03	58.7 ± 3.6	426 ± 85	188 ± 28

**Table 2 polymers-14-05254-t002:** Mechanical properties of PLA and PPCOG non-woven mats in dry and wet state.

Sample	EA-SBS		ES	
	Dry	Wet	Dry	Wet
Elastic modulus, MPa	25 ± 4	17 ± 3	72 ± 9	54 ± 6
Tensile strength, MPa	1.15 ± 0.21	1.18 ± 0.03	3.9 ± 0.3	2.9 ± 0.1
Elongation at break, %	7.7 ± 0.3	18 ± 3	14 ± 4	16 ± 2

## Data Availability

The data presented in this study are available on request from the corresponding author.

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
