# Peer review of "Electrospinning vs. Electro-Assisted Solution Blow Spinning for Fabrication of Fibrous Scaffolds for Tissue Engineering"

_polymers, 2022, doi:10.3390/polym14235254_

Round 1

Reviewer 1 Report

In this paper, the biodegradable polymer fiber non-woven materials prepared by different methods are studied, and the influencing factors on the morphology and properties of fiber materials are explored. However, there are some critical issues the authors should resolve during the revision process as follows:

1.      It is suggested to cite relevant papers published in recent three years, such as: Blow spinning: AirBlowingAssisted Coaxial Electrospinning toward High Productivity of Core/Sheath and Hollow Fibers; Structural design: Structural design toward functional materials by electrospinning: A review; Core effect on mechanical properties of one dimensional electrospun core-sheath composite fibers

2.      The formatting between numbers and units in whole manuscript is inconsistent. Authors are suggested to carefully recheck the whole manuscript to revise the grammar issues.

3.      More information on the raw materials should be provided, such as molecular weight, purity, etc.

4.      The characterization of the mechanical properties did not appear in the manuscript, and the authors considered whether it was necessary to supplement the mechanical properties testing of fibrous non-woven materials used as scaffolds. If necessary, it is recommended to supplement the mechanical properties.

5.      The scale bar for SEM should be remade to ensure the resolution and readability.

6.      The applications of electrospun nonwovens should be provided in the introduction section to present the importance of electrospinning. Some recent articles on the applications can be considered, such as: food packaging (Nanomaterials 10 (1), 150, 2020); EMI shielding (Electrospun fibrous materials and their applications for electromagnetic interference shielding: A review); metal-air batteries (Materials & Design 214, 110406, 2022); smart textile (A review of smart electrospun fibers toward textiles); etc.

7.      In order to make the data more convincing, it is suggested to add error bars in some of the pictures in the manuscript.

8.      The author thinks that “The lower value of initial contact angle and its faster decrease in time could be explained by lower diameter of electrospun fibers.” The corresponding proof is not given in the manuscript. It is suggested that the author make reasonable explanation or introduce relevant literature to support it.

9.      The present writing style of conclusions is unusual. It is suggested to write this section in one paragraph logically.

Author Response

In this paper, the biodegradable polymer fiber non-woven materials prepared by different methods are studied, and the influencing factors on the morphology and properties of fiber materials are explored. However, there are some critical issues the authors should resolve during the revision process as follows:

  1. It is suggested to cite relevant papers published in recent three years, such as: Blow spinning: Air‐Blowing‐Assisted Coaxial Electrospinning toward High Productivity of Core/Sheath and Hollow Fibers; Structural design: Structural design toward functional materials by electrospinning: A review; Core effect on mechanical properties of one dimensional electrospun core-sheath composite fibers

Answer: Thank you for your suggestion! We updated the Introduction section.

  1. The formatting between numbers and units in whole manuscript is inconsistent. Authors are suggested to carefully recheck the whole manuscript to revise the grammar issues.

Answer: The manuscript was re-checked.

  1. More information on the raw materials should be provided, such as molecular weight, purity, etc.

Answer: The information was added.

  1. The characterization of the mechanical properties did not appear in the manuscript, and the authors considered whether it was necessary to supplement the mechanical properties testing of fibrous non-woven materials used as scaffolds. If necessary, it is recommended to supplement the mechanical properties.

Answer: Thank you for the suggestion! We carried out the mechanical testing and added this information within the updated version of the manuscript.

  1. The scale bar for SEM should be remade to ensure the resolution and readability.

Answer: The scale bar was remade.

  1. The applications of electrospun nonwovens should be provided in the introduction section to present the importance of electrospinning. Some recent articles on the applications can be considered, such as: food packaging (Nanomaterials 10 (1), 150, 2020); EMI shielding (Electrospun fibrous materials and their applications for electromagnetic interference shielding: A review); metal-air batteries (Materials & Design 214, 110406, 2022); smart textile (A review of smart electrospun fibers toward textiles); etc.

Answer: Thank you for the suggestions! The Introduction section was updated.

  1. In order to make the data more convincing, it is suggested to add error bars in some of the pictures in the manuscript.

Answer: Error bars are added. In some cases (Figure 4), the errors are very small to be clearly visible. We updated this Figure to ensure the visibility. The contact angles (Figure 2) are shown as average values with the standard deviation ± 1 degree. We added this information within the experimental section to not overload the graph.

  1. The author thinks that “The lower value of initial contact angle and its faster decrease in time could be explained by lower diameter of electrospun fibers.” The corresponding proof is not given in the manuscript. It is suggested that the author make reasonable explanation or introduce relevant literature to support it.

Answer: Thank you for the comment! The lower value of initial contact angle and its faster decrease in time could be explained by different morphology of the electrospun fibers. The EA-SBS material consists of fibers having the higher surface roughness and unevenness in diameter compared to ES, which could provoke to trap more air causing more water repulsion [Wang, D.; Yue, Y.; Wang, Q.; Cheng, W.; Han, G. Preparation of cellulose acetate-polyacrylonitrile composite nanofibers by multi-fluid mixing electrospinning method: Morphology, wettability, and mechanical properties. Appl. Surf. Sci. 2020, 510, 145462, doi:10.1016/j.apsusc.2020.145462].

  1. The present writing style of conclusions is unusual. It is suggested to write this section in one paragraph logically.

Answer: The conclusions were merged in one paragraph.

Reviewer 2 Report

In the manuscript “Electrospinning vs. solution blow spinning for fabrication of fibrous scaffolds for tissue engineering” submitted to “Polymers” for publication, the authors have developed various fibrous scaffolds from multicomponent polymeric system containing biodegradable synthetic (polylactide, polycaprolactone) and natural (gelatin, chitosan) components using different methods of non-woven mats fabrication: electrospinning, solution blow spinning and electro-assisted solution blow spinning.

The manuscript needs some major improvements; there are a few suggestions that authors may consider to improve it further:

The use of English language is reasonable, however, there are a number of punctuation and grammatical errors; that should be corrected and rephrased using academic English for a better flow of text for reader.

The main issue is the lack of novelty, the authors need to justify the new information as a number of si ilar studies on similar materials are already there.

There is no information about the biodegradation of the prepared materials? How would it be controlled and characterized? Is this aspect not included as a part of this study?

Please further discuss limitations of the study.

Introduction 1st paragraphs, authors used solid claim about the electrospinning, which as hard to accept; such as the main types of materials, the most popular etc. also, author may benefit from these articles to clarify and further expand the concept of electrospinning of polymers:

https://www.mdpi.com/1422-0067/19/2/407

https://www.mdpi.com/1996-1944/9/2/73

Please justify the selection of natural polymers: as some other polymers are widely used (such as silk) but not used in this study?

The authors should included more studies in the discussion section and comparing findings.

Please further discuss limitations of the study. 

Author Response

In the manuscript “Electrospinning vs. solution blow spinning for fabrication of fibrous scaffolds for tissue engineering” submitted to “Polymers” for publication, the authors have developed various fibrous scaffolds from multicomponent polymeric system containing biodegradable synthetic (polylactide, polycaprolactone) and natural (gelatin, chitosan) components using different methods of non-woven mats fabrication: electrospinning, solution blow spinning and electro-assisted solution blow spinning.

The manuscript needs some major improvements; there are a few suggestions that authors may consider to improve it further:

The use of English language is reasonable, however, there are a number of punctuation and grammatical errors; that should be corrected and rephrased using academic English for a better flow of text for reader.

Answer: Thank you for the comment! We revised the manuscript.

The main issue is the lack of novelty, the authors need to justify the new information as a number of si ilar studies on similar materials are already there.

Answer: We updated the Introduction section. EA-SBS is not so widely used technique for fabrication of scaffolds for tissue engineering in contrast to electrospinning. There are only a few works on cell culture onto EA-SBS scaffolds. This work was aimed at comparison of cell adhesion and growth onto the fibrous materials fabricated using EA-SBS and ES techniques.

There is no information about the biodegradation of the prepared materials? How would it be controlled and characterized? Is this aspect not included as a part of this study?

Answer: Thank you for the question! We used the multicomponent system consisting of polymers, which are biodegradable (oligo/polylactide, polycaprolactone, chitosan and gelatin). We are planning to carry out in vivo experiments on biocompatibility and biodegradability of these fibrous materials further.

Introduction 1st paragraphs, authors used solid claim about the electrospinning, which as hard to accept; such as the main types of materials, the most popular etc. also, author may benefit from these articles to clarify and further expand the concept of electrospinning of polymers:

https://www.mdpi.com/1422-0067/19/2/407
https://www.mdpi.com/1996-1944/9/2/73

Answer: We agree with the Referee. This paragraph was modified.

Please justify the selection of natural polymers: as some other polymers are widely used (such as silk) but not used in this study?

Answer: We used the complex multicomponent polymeric system, which was previously studied and characterized. This work was aimed at comparison of the fibrous materials fabricated using different techniques.

The authors should included more studies in the discussion section and comparing findings.

Answer: The discussion section was modified.

Please further discuss limitations of the study. 

Answer: The high cell attachment and proliferation onto EA-SBS fibrous materials show them as promising type of scaffolds for tissue engineering, but it’s just a beginning of their study as scaffolds. The future in vivo experiments on biocompatibility and biodegradability is planned.

Round 2

Reviewer 1 Report

Authors seemed made revisions according to the comments. However, all the reponses to the comments are general answers. It is difficult to find the specific revisions to each comment. Authors are suggested to make a new response to previous comments, including the point-to-point responses and the corresponding revisions in the main manuscript.  

Author Response

Authors seemed made revisions according to the comments. However, all the reponses to the comments are general answers. It is difficult to find the specific revisions to each comment. Authors are suggested to make a new response to previous comments, including the point-to-point responses and the corresponding revisions in the main manuscript.

Answer: We apologize for the inconvenience! Please, find point-to-point responses to the previous comments:

1. It is suggested to cite relevant papers published in recent three years, such as: Blow spinning: Air‐Blowing‐Assisted Coaxial Electrospinning toward High Productivity of Core/Sheath and Hollow Fibers; Structural design: Structural design toward functional materials by electrospinning: A review; Core effect on mechanical properties of one dimensional electrospun core-sheath composite fibers

Answer: We added several relevant papers (including ones published in recent three years) in the Introduction section:

  1. Sameen, D.E.; Ahmed, S.; Lu, R.; Li, R.; Dai, J.; Qin, W.; Zhang, Q.; Li, S.; Liu, Y. Electrospun nanofibers food packaging: trends and applications in food systems. Crit. Rev. Food Sci. Nutr. 2022, 62, 6238–6251, doi:10.1080/10408398.2021.1899128.
  2. Feng, X.; Li, J.; Zhang, X.; Liu, T.; Ding, J.; Chen, X. Electrospun polymer micro/nanofibers as pharmaceutical repositories for healthcare. J. Control. Release 2019, 302, 19–41, doi:10.1016/j.jconrel.2019.03.020.
  3. Arida, I.A.; Ali, I.H.; Nasr, M.; El-Sherbiny, I.M. Electrospun polymer-based nanofiber scaffolds for skin regeneration. J. Drug Deliv. Sci. Technol. 2021, 64, 102623, doi:10.1016/j.jddst.2021.102623.
  4. Maleki, H.; Azimi, B.; Ismaeilimoghadam, S.; Danti, S. Poly(lactic acid)-Based Electrospun Fibrous Structures for Biomedical Applications. Appl. Sci. 2022, 12, 3192, doi:10.3390/app12063192.
  5. Li, Q.; Wang, X.; Lou, X.; Yuan, H.; Tu, H.; Li, B.; Zhang, Y. Genipin-crosslinked electrospun chitosan nanofibers: Determination of crosslinking conditions and evaluation of cytocompatibility. Carbohydr. Polym. 2015, 130, 166–174, doi:10.1016/j.carbpol.2015.05.039.
  6. Qasim, S.; Zafar, M.; Najeeb, S.; Khurshid, Z.; Shah, A.; Husain, S.; Rehman, I. Electrospinning of Chitosan-Based Solutions for Tissue Engineering and Regenerative Medicine. Int. J. Mol. Sci. 2018, 19, 407, doi:10.3390/ijms19020407.
  7. Duan, G.; Greiner, A. Air‐Blowing‐Assisted Coaxial Electrospinning toward High Productivity of Core/Sheath and Hollow Fibers. Macromol. Mater. Eng. 2019, 304, 1800669, doi:10.1002/mame.201800669.
  8. Lauricella, M.; Succi, S.; Zussman, E.; Pisignano, D.; Yarin, A.L. Models of polymer solutions in electrified jets and solution blowing. Rev. Mod. Phys. 2020, 92, 035004, doi:10.1103/RevModPhys.92.035004.
  9. Choi, M.; Kim, J. Development of Coaxial Air-blown Electrospinning Process for Manufacturing Non-woven Nanofiber. II. Intelligent Modeling. Fibers Polym. 2019, 20, 1883–1892, doi:10.1007/s12221-019-1094-z.
  10. Cao, L.; Liu, Q.; Ren, J.; Chen, W.; Pei, Y.; Kaplan, D.L.; Ling, S. Electro‐Blown Spun Silk/Graphene Nanoionotronic Skin for Multifunctional Fire Protection and Alarm. Adv. Mater. 2021, 33, 2102500, doi:10.1002/adma.202102500.
  11. Al Rai, A.; Stojanovska, E.; Fidan, G.; Yetgin, E.; Polat, Y.; Kilic, A.; Demir, A.; Yilmaz, S. Structure and performance of electroblown PVDF‐based nanofibrous electret filters. Polym. Eng. Sci. 2020, 60, 1186–1193, doi:10.1002/pen.25372.
  12. Sadeghzadeh, A.; Bazgir, S.; Shirazi, M.M.A. Fabrication and characterization of a novel hydrophobic polystyrene membrane using electroblowing technique for desalination by direct contact membrane distillation. Sep. Purif. Technol. 2020, 239, 116498, doi:10.1016/j.seppur.2019.116498.
  13. Aminyan, R.; Bazgir, S. Fabrication and characterization of nanofibrous polyacrylic acid superabsorbent using gas-assisted electrospinning technique. React. Funct. Polym. 2019, 141, 133–144, doi:10.1016/j.reactfunctpolym.2019.05.012.
  14. Holopainen, J.; Ritala, M. Rapid production of bioactive hydroxyapatite fibers via electroblowing. J. Eur. Ceram. Soc. 2016, 36, 3219–3224, doi:10.1016/j.jeurceramsoc.2016.05.011.
  15. Dias, Y.J.; Robles, J.R.; Sinha-Ray, S.; Abiade, J.; Pourdeyhimi, B.; Niemczyk-Soczynska, B.; Kolbuk, D.; Sajkiewicz, P.; Yarin, A.L. Solution-Blown Poly(hydroxybutyrate) and ε-Poly- <scp>l</scp> -lysine Submicro- and Microfiber-Based Sustainable Nonwovens with Antimicrobial Activity for Single-Use Applications. ACS Biomater. Sci. Eng. 2021, 7, 3980–3992, doi:10.1021/acsbiomaterials.1c00594.
  16. Rampichová, M.; Chvojka, J.; Jenčová, V.; Kubíková, T.; Tonar, Z.; Erben, J.; Buzgo, M.; Daňková, J.; Litvinec, A.; Vocetková, K.; et al. The combination of nanofibrous and microfibrous materials for enhancement of cell infiltration and in vivo bone tissue formation. Biomed. Mater. 2018, 13, doi:10.1088/1748-605X/aa9717.

2. The formatting between numbers and units in whole manuscript is inconsistent. Authors are suggested to carefully recheck the whole manuscript to revise the grammar issues.

Answer: The manuscript was re-checked and we corrected all found inconsistencies and grammar issues. All these corrections are made using “Track changes” mode of Word Office. There is a considerable number of changes, so it would be complicated to mention all of them.

3. More information on the raw materials should be provided, such as molecular weight, purity, etc.

Answer: The information was added as the first paragraph of the Experimental section:

“Polylactide (PLA) (Natureworks 4043D, Minnetonka, MN, USA) with molecular weight (Mw) of 100 kDa; polycaprolactone (PCL) with Mw of 67 kDa and gelatin (Chimmed, Russia) were used as received. Chitosan with Mw of 80 kDa and degree of deacetylation of 0.89 as well as oligo(l-lactide) with Mw of 5 kDa from l-lactic acids (Panreac, Barcelona, Spain) were synthesized in Enikolopov Institute of Synthetic Polymeric Materials.”

4. The characterization of the mechanical properties did not appear in the manuscript, and the authors considered whether it was necessary to supplement the mechanical properties testing of fibrous non-woven materials used as scaffolds. If necessary, it is recommended to supplement the mechanical properties.

Answer: Thank you for the suggestion! We carried out the mechanical testing and added this information within the updated version of the manuscript.

            Within the Section 2.2. (Experimental section):

“Mechanical properties of the fibrous samples were tested in dry and wet states using Mach-1 v500csst Micromechanical Testing System (Biomomentum Inc., Laval, Canada). Testing of the wet samples was conducted after their incubation in mQ water for 2 days during constant lateral stirring at 200-300 rpm at RT. The tensile strength, elongation at break and elasticity modulus (Young’s modulus) were measured during uniaxial tension of at least three dog-bone shape cutted samples (13x5 mm gauge dimensions, the thickness was 0.17±0.1 mm for EA-SBS and 0.19±0.1 mm for ES samples, respectively) and reported as mean ± standard deviation values.”

Within the Section 3.1.2. (Results):

“The samples were also mechanically tested not only in dry state, but in wet state to better represent the material behavior within the cell culture conditions. Macrophotos of the samples at the beginning and at the end of the deformation are shown in Figure 3, while obtained results of the mechanical testing are summarized in Table 2.

Figure 3

Table 2

ES samples had higher elastic modulus, tensile strength and elongation at break in the dry state. Incubation of the samples in water before the testing caused swelling of the hydrophilic biopolymers which led to some degree of softening (a decrease in elastic modulus and tensile strength), at the same time an increase in elongation at break was observed for the EA-SBS sample. In overall, the samples demonstrated mechanical properties expected for the similar polymer systems and are in a range of ones of connective tissues [29,35].”

Within the Discussion section:

“Mechanical properties of the fibrous mats formed by the ES method are higher than that of scaffolds formed by the EA-SBS method. Since the ES scaffolds are formed mainly by submicron fibers, while the average fiber diameter in EA-SBS scaffolds reaches tens of microns (Table 2), the specific load onto each fiber is less for scaffolds formed by the ES method, which determines their greater absolute strength. Moreover, SEM images (Figure 1) show that the electrospin fibers have a regular cylindrical shape, while the EA-SBS-formed fibers have a more complex morphology and macrodefects in the form of drops. Therefore, a more perfect fiber structure of ES scaffolds contributes to their better mechanical properties.”

5. The scale bar for SEM should be remade to ensure the resolution and readability.

Answer: The scale bar was remade. Please, see Figure 1.

6. The applications of electrospun nonwovens should be provided in the introduction section to present the importance of electrospinning. Some recent articles on the applications can be considered, such as: food packaging (Nanomaterials 10 (1), 150, 2020); EMI shielding (Electrospun fibrous materials and their applications for electromagnetic interference shielding: A review); metal-air batteries (Materials & Design 214, 110406, 2022); smart textile (A review of smart electrospun fibers toward textiles); etc.

Answer: Thank you for the suggestions! The Introduction section was updated and the recent articles on the applications of electrospun nonwovens were added:

“3. Sameen, D.E.; Ahmed, S.; Lu, R.; Li, R.; Dai, J.; Qin, W.; Zhang, Q.; Li, S.; Liu, Y. Electrospun nanofibers food packaging: trends and applications in food systems. Crit. Rev. Food Sci. Nutr. 2022, 62, 6238–6251, doi:10.1080/10408398.2021.1899128.

  1. Feng, X.; Li, J.; Zhang, X.; Liu, T.; Ding, J.; Chen, X. Electrospun polymer micro/nanofibers as pharmaceutical repositories for healthcare. J. Control. Release 2019, 302, 19–41, doi:10.1016/j.jconrel.2019.03.020.
  2. Arida, I.A.; Ali, I.H.; Nasr, M.; El-Sherbiny, I.M. Electrospun polymer-based nanofiber scaffolds for skin regeneration. J. Drug Deliv. Sci. Technol. 2021, 64, 102623, doi:10.1016/j.jddst.2021.102623.”

7. In order to make the data more convincing, it is suggested to add error bars in some of the pictures in the manuscript.

Answer: Error bars are added. In some cases (Figure 4), the errors have been shown within the initial version of the manuscript, but very small to be clearly visible. We updated this Figure 4 to ensure the visibility. Error bars in Figure 3 are presented and are visible. The contact angles (Figure 2) are shown as average values with the standard deviation ± 1 degree. We added this information within the experimental section to not overload the graph. The sentence was added within the Section 2.2.: “The contact angles are reported as the average of at least three measurements; the standard deviation for contact angles was ± 1 degree.”

8. The author thinks that “The lower value of initial contact angle and its faster decrease in time could be explained by lower diameter of electrospun fibers.” The corresponding proof is not given in the manuscript. It is suggested that the author make reasonable explanation or introduce relevant literature to support it.

Answer: Thank you for the comment! The explanation was corrected and extended:

“The lower value of initial contact angle and its faster decrease in time could be explained by different morphology of the electrospun fibers. The EA-SBS material consists of fibers having the higher surface roughness and unevenness in diameter compared to ES, which could provoke to trap more air causing more water repulsion [Wang, D.; Yue, Y.; Wang, Q.; Cheng, W.; Han, G. Preparation of cellulose acetate-polyacrylonitrile composite nanofibers by multi-fluid mixing electrospinning method: Morphology, wettability, and mechanical properties. Appl. Surf. Sci. 2020, 510, 145462, doi:10.1016/j.apsusc.2020.145462]”

9. The present writing style of conclusions is unusual. It is suggested to write this section in one paragraph logically.

Answer: The conclusion were merged in one paragraph and written as follows:

“Electro-assisted solution blow spinning technique allows fabricating non-woven fibrous materials consisting of multifilament fibers in contrast to monofilament electrospun fibers. Using the same concentration of the spinning solution the EA-SBS led to formation of fibers with the bigger mean diameter size than ones of the fibers produced by ES. The diameter size distribution was wider in the case of using EA-SBS technique. The multifilament structure of the fibrous materials produced via EA-SBS provided the better mammalian cell adhesion and growth than electrospun fibers. This phenomenon could be caused by more rough multifilament fiber surface. The found superiority of multifilament fibrous materials in contrast to monofilament ones in terms of cell attachment and proliferation makes them a highly promising type of scaffolds for tissue engineering.”

Reviewer 2 Report

Many thanks for revising the manuscript.

Author Response

Thank you very much for your time and the valuable advice!